# Isolation and diversity of sediment bacteria in the hypersaline aiding lake, China

**Tong-Wei Guan** 📷 *, **Yi-Jin Lin, Meng-Ying Ou, Ke-Bao Chen**

Institute of Microbiology, Xihua University, Chengdu, P.R. China

* guantongweily@163.com

**Data Availability Statement:** All relevant data are within the paper. The sequences of the bacterial isolates reported in this study have been deposited to GenBank (Accession no. MK818765-MK818834, MK296404).

## Abstract

Halophiles are relatively unexplored as potential sources of novel species. However, little is known about the culturable bacterial diversity thrive in hypersaline lakes. In this work, a total of 343 bacteria from sediment samples of Aiding Lake, China, were isolated using nine different media supplemented with 5% or 15% (w/v) NaCl. The number of species and genera of bacteria recovered from the different media varied, indicating the need to optimize the isolation conditions. The results showed an unexpected level of bacterial diversity, with four phyla (*Actinobacteria*, *Firmicutes*, *Proteobacteria*, and *Rhodothermaeota*), fourteen orders (*Actinopolysporales*, *Alteromonadales*, *Bacillales*, *Balneolales*, *Chromatiales*, *Glycomycetales*, *Jiangellales*, *Micrococcales*, *Micromonosporales*, *Oceanospirillales*, *Pseudonocardiales*, *Rhizobiales*, *Streptomycetales*, and *Streptosporangiales*), including 17 families, 43 genera (including two novel genera), and 71 species (including four novel species). The predominant phyla included Actinobacteria and Firmicutes and the predominant genera included *Actinopolyspora*, *Gracilibacillus*, *Halomonas*, *Nocardiopsis*, and *Streptomyces*. To our knowledge, this is the first time that members of phylum *Rhodothermaeota* were identified in sediment samples from a salt lake.

## 1 Introduction

Halophiles thrive in hypersaline niches and have potential applications in biotechnology [1, 2]. Microbial diversity in most hypersaline environments is often studied using culture-dependent and -independent methods [3–7]. Previous studies have shown that the taxonomic diversity of microbial populations in terrestrial saline and hypersaline environments is relatively low [8, 9]. Halophilic microbial communities vary with season [10], and in general, microbial diversity decreases with increased salinity [11, 12]. Hypersaline lakes are considered extreme environments for microbial life. A variety of salt lakes have been surveyed for bacterial diversity such as Chaka Lake in China, Chott El Jerid Lake in Tunisia, Meyghan Lake in Iran, Keke Lake in China, and Great Salt Lake in the United States [5, 13–16]. In addition, groups of novel halophilic or halotolerant bacteria in salt lakes have been described using culture-dependent methods: *Actinopolyspora lacussalsi* sp. nov., *Amycolatopsis halophila* sp. nov., *Brevibacterium salitolerans* sp. nov., *Halomonas xiaochaidanensis* sp. nov., *Paracoccus halotolerans* sp. nov., and *Salibacterium nitratireducens* sp. nov. of phyla *Actinobacteria*, *Firmicutes*, or

**Funding:** The National Natural Science Foundation of China (Project No. 30660005) and Office of education in Sichuan Province, China (Project Nos. 13205688, 13ZB0024) supported this study. Chunhui Project for Ministry of Education of China (Project No. 191649).

**Competing interests:** The authors have declared that no competing interests exist.

*Proteobacteria* [17–22]. Despite these previous studies, our understanding of bacterial diversity in hypersaline lakes remains limited, particularly in athalassohaline lakes at low elevations. Aiding Lake represents an ideal site for studying halophilic or halotolerant bacteria in a hypersaline lake. The salt lake is located on the Turpan Basin and surround by the Gobi desert in Xinjiang province, China. In fact, our current understanding of the bacterial diversity in the Aiding Lake using a culture-dependent method is limited. To our knowledge, this is the first attempt to comprehensively characterize the bacterial diversity in dry salt lake sediments. The aim of the study was to investigate the bacterial diversity and to mine novel bacterial species from Aiding Lake.

## 2 Materials and methods

### 2.1 Site description and sample collection

Aiding Lake is a dry salt lake located on the Turpan Basin in Xinjiang Province, China (S1 Fig), with an elevation of 154 m–293 m below sea level. Aiding Lake covers an area of about 60 km$^2$ and is a closed ecosystem, without the influx of perennial rivers. Three soil samples, namely, S1 (89˚21′98″E, 42˚40′42″N), S2 (89˚16′6″E, 42˚38′55″N), and S3 (89˚20'26″E, 42˚41'53″N) were collected from the lake sediments, respectively. Soil samples temperature are 23.6˚C-25.1˚C. Three sediment samples at different sites were collected from a same depth of 1 to 30 cm in mid-July of 2012. The distance between two sample points was greater than 5 km. Samples were stored at 4˚C in the field and immediately transported to the laboratory. The pH was measured with portable meters after the sediments were resuspended in distilled water. The concentrations of major cations and trace elements in the dissolved sediments were measured according to Yakimov et al. (2002) [23].

### 2.2 Isolation of microorganisms

Three sediment samples were selected for cultivation of bacteria. To isolate halophilic and/or halotolerant bacteria, the sediments (10 g wet weight) were dispersed into 90 mL of sterilized NaCl brine (5% or 15%, w/v) and incubated at 37˚C for 60 min with shaking at 200 rpm. The resulting slurry was then serially diluted with sterilized NaCl brine (5% or 15%, w/v). Aliquots (0.1 mL) of each dilution were spread onto Petri dishes using nine media (Table 1) for the isolation of bacteria. The no. of colonies on each kind of medium plate was calculated by three repeats. All agar plates were supplemented with 5% or 15% (w/v) NaCl. To suppress the growth of nonbacterial fungi, the solidified media were supplemented with nystatin (50 mg·L$^{-1}$). The Petri dishes were incubated at 37˚C for one to six weeks. Based on size and color, colonies were picked and further purified on inorganic salts-starch agar [24] or TSA supplemented with 5% or 15% (w/v) NaCl, and as glycerol suspension (20%, v/v) at -20˚C or as lyophilized cells for long-term storage at -4˚C.

### 2.3 Identification of bacteria

Isolated strains were subjected to 16S rRNA gene sequence analysis for precise genus and species identification. Genomic DNA was extracted from each isolate, and the 16S rRNA gene sequence was amplified as described by Li et al. (2007) [25] with primers PA (5'-CAGAGTT TGATCCTGGCT-3') and PB (5'-AGG AGGTGATCCAGCCGC A-3'), or the primers 27F (5'-AGAGTTTGATCMTGGCTCAG-3') and 1492R (5'-GGTTACCTTGTT ACGACTT-3'). PCR products were purified using a PCR purification kit (Sangon, Shanghai, China). The almost-complete 16S rRNA gene sequence (about 1450bp) of isolated strains was obtained. Multiple alignments with sequences of the most closely related recognized species and

**Table 1. Compositions of the nine different media used for the isolation of bacteria from Aiding Lake samples.**

| Medium | Composition | Reference |
|---|---|---|
| A | Inorganic salts-starch agar (ISP 4) | Shirling and Gottlieb 1966 |
| B | Casein hydrolysate acid-starch agar: starch 5.0 g, casein hydrolysate acid 0.5 g, $KNO_3$ 0.5 g, Aspartic acid 0.1g, $CaCO_3$, 0.3 g, $K_2HPO_4$ 0.5 g, $MgCl_2$ 0.2 g, $FeSO_4·7H_2O$ 10 mg, agar 18 g | This study |
| C | Microcrystalline cellulose-proline agar: microcrystalline cellulose 2.0 g, proline 0.5 g, arginine 0.1 g, KCl 10 g, $(NH_4)_2SO_4$ 1.0 g, $K_2HPO_4$ 0.2 g, $CaCO_3$ 0.02 g, $MgSO_4 · 7H_2O$ 2 g, $FeSO_4 ·7H_2O$ 10 mg, $MnCl_2·4H_2O$ 1mg, agar 18 g | This study |
| D | Glycerin-asparagine agar: glycerin 3g, asparagine 1g, $C_3H_3NaO_3$ 0.5g, $MgSO_4·7H_2O$ 2 g, $FeSO_4·7H_2O$ 10 mg, $ZnSO_4·7H_2O$ 1 mg, VB1 0.1 mg, VB6 0.05 mg, biotin 0.2mg, agar 18 g | This study |
| E | Yeast extract-casamino acids agar: yeast extract 3g, Casamino acids 2g, Sodium glutamate 1g, Trisodium citrate 1g, $MgSO_4·7H_2O$ 5g, $CaCl_2·2H_2O$ 1g, KCl 3g, $FeCl_2·4H_2O$ 0.2mg, $MnCl_2·4H_2O$ 0.2mg, agar, 18 g | This study |
| F | Stachyose tetrahydrate-Alanine agar: stachyose tetrahydrate 5g, alanine 2g, $KNO_3$ 0.2g, $CaCO_3$ 0.02g, $MgSO_4.7H_2O$ 0.05g, KCl 20g, $FeSO_4·7H_2O$ 10 mg, $MgCl_2$ 20g, agar 18 g | This study |
| G | Microcrystalline cellulose-sorbitol agar: microcrystalline cellulose 10g, sorbitol 2g, Beta-Cyclodextrin 1g, $MgSO_4·7H_2O$ 0.1g, $CaCO_3$ 0.5g, $FeSO_4$ 0.01g, KCl 20g, $MgCl_2$ 10g, agar 18 g | This study |
| H | Yeast extract-fish peptone agar: yeast extract 1g, fish peptone 0.5g, $NH_4Cl$ 0.5g, $MgSO_4.7H_2O$ 20g, $MgCl_2·6H_2O$ 15g, KCl 5g, sodium pyruvate 1g, $K_2HPO_4$ 0.3g, $CaCl_2·2H_2O$ 0.2g, agar 18 g | This study |
| I | Yeast extract-glycerin agar: yeast extract 10g, Glycerin 0.5g, peptone 0.5g, $(NH_4)NO_3$ 0.1g, $MgCl_2$ 5g, $Na_2SO_4$ 3g, yeast KCl 1g, $NaHCO_3$ 2g, KBr 0.05g, $SrCl_2$ 0.01g, $Na_2SiO_3$ 0.001g, agar 18 g | This study |

calculations of levels of sequence similarity were conducted using EzBioCloud server [26]. Phylogenetic analysis was performed using the software package MEGA version 6.0 [27]. Phylogenetic trees were constructed according to the neighbor-joining method [28]. Evolutionary distance matrices were generated as described by Kimura (1980) [29]. The topology of the phylogenetic tree was evaluated using the bootstrap resampling method of Felsenstein (1985) [30] with 1000 replicates. DNA-DNA relatedness values were determined using the fluorometric microwell method [31]. The identities of these organisms were determined based on nearly full-length 16S rRNA gene sequence analysis. The sequences of 100% identity were clustered into one species.

### 2.4 Spearman correlation analysis

Pearson's test was performed to reveal the correlations between physicochemical properties and bacterial genera using SPSS Statistics19.0.

### 2.5 Nucleotide sequence accession numbers

The sequences of the bacterial isolates reported in this study have been deposited to GenBank (Accession no. MK818765- MK818834, MK296404).

## 3 Results

### 3.1 Sediment geochemistry

Physicochemical parameters were distinct among the three sediment samples (Table 2). In the S1, S2, and S3 samples, $Na^+$ concentrations ranged from 26.37 g/Kg to 83.93 g/Kg and $Cl^-$

**Table 2. Physicochemical properties of the sediments from the three sample sites in Aiding Lake.**

| Site | pH | Ion concentration (g Kg⁻¹) | | | | | | | | |
|------|-----|----------|----------|----------|----------|----------|----------|----------|----------|----------|
| | | $Ca^{2+}$ | $Mg^{2+}$ | $Fe^{2+}$ | $Na^+$ | $K^+$ | $Mn^{2+}$ | $Cl^-$ | $SO_4^{2-}$ | $HCO_3^-$ |
| S1 | 8.3 | 6.54 | 1.93 | 63ppm | 83.93 | 0.33 | 23ppm | 389.11 | 85.25 | 0.06 |
| S2 | 8.1 | 0.24 | 2.43 | 102ppm | 38.29 | 0.25 | 9ppm | 45.38 | 6.42 | 0.13 |
| S3 | 7.6 | 4.25 | 0.61 | 82ppm | 26.37 | 0.16 | 2ppm | 33.28 | 23.52 | 0.41 |

ppm, parts per million.

concentrations ranged from 33.28 g/Kg to 389.11 g/Kg, which are typical of chloride-type environments. The pH of the sediment samples ranged from 7.6 to 8.3, indicating a slightly alkaline environment. In sample S3, ionic composition (e.g., $Mg^{2+}$, $Na^+$, $K^+$, $Mn^{2+}$, and $Cl^-$) was significantly lower than the other samples. The other physicochemical properties of the samples are presented in Table 2.

## 3.2 Diversity of sediment bacteria

According to the analysis of sequencing results, many of isolated strains had exactly the same 16S rRNA sequence. 343 isolated strains belong to 71 different bacterial species after merging the duplicated strains (Table 3). The percentages of 16S rRNA gene sequence similarities (91.14% to 100%) of these isolates to the closest type strains are presented in Table 3. As the results indicated, most of the strains exhibited > 97% similarity to other published type species. While the similarity of three strains (ADL013, ADL014, and ADL023) were less than 97%, which represented three different novel species (Table 3). It is generally accepted that organisms displaying 16S rRNA gene sequence similarity values of 97% or less belong to different species [32]. For example, strain ADL014 shared 96.51% similarity with *Anaerobacillus alkalidiazotrophicus* F01CH1-61-65, and DNA-DNA hybridization experiments from the two strains revealed that levels of DNA-DNA relatedness were 36.8± 4.3%. Sequence analysis indicated that strain ADL014 formed a distinct lineage within the genus *Anaerobacillus* and always had the closest phylogenetic affinity to members of the genus *Anaerobacillus* (Fig 1). Phylogenetic reconstruction also indicated that strain ADL014 could represent a novel species.

These halophilic or halotolerant bacteria were compared to those deposited in the public database (EzBioCloud, https://www.ezbiocloud.net/identify). The bacteria isolated in this study displayed considerable diversity. The predominant phyla were *Firmicutes* (149 strains, 43.4%) and *Actinobacteria* (121 strains, 35.3%). The other bacterial isolates belonged to phyla *Rhodothermaeota* (5 strains, 1.5%) and *Proteobacteria* (68 strains, 19.8%). The isolates were distributed among 14 orders, namely, *Actinopolysporales* (26 strains), *Alteromonadales* (8 strains), *Bacillaceae* (149 strains), *Balneolales* (5 strains), *Chromatiales* (12 strains), *Glycomycetales* (2 strains), *Jiangellales* (3 strains), *Micrococcales* (8 strains), *Micromonosporineae* (5 strains), *Oceanospirillales* (45 strains), *Pseudonocardiales* (11 strains), *Rhizobiales* (3 strains), *Streptomycetales* (27 strains), and *Streptosporangiales* (26 strains), including 41 known genera (Tables 3 and 4). Other organisms (ADL013 and ADL023) could not be accurately identified to the genus level because of the lower homology (Table 3). Strain ADL013 exhibited 94.71% similarity to the 16S rRNA gene sequence of *Alteribacillus persepolensis* HS136, and the hybridization values of 11.3±2.1% to each other; and strain ADL023 exhibited 91.14% similarity to the 16S rRNA gene sequence of *Caldalkalibacillus uzonensis* JW/WZ-YB58, and the hybridization values of 8.3±1.7% to each other. Phylogenetic analysis also showed that strain ADL013 and ADL023 can be distinguished from representatives of genera in the family *Bacillaceae*, and two strains formed a distinct lineage within family *Bacillaceae*, respectively (Fig 2). Meanwhile,

**Table 3. Bacteria isolated using two different salinity from sediments of Aiding Lake, with the similarity values for 16S rRNA gene sequences.**

| Isolate | Salinity | Closest cultivated species (GenBank accession no.) | Similarity (%) | No. of isolates |
|---|---|---|---|---|
| ADL001 | 15% | *Actinopolyspora alba* YIM 90480 (GQ480940) | 98.50 | 3 |
| ADL003 | 15% | *Actinopolyspora mortivallis* DSM 44261(NR_043996.1) | 98.80 | 1 |
| ADL004 | 15% | *Actinopolyspora halophila* DSM 43834 (AQUI01000002) | 100 | 9 |
| ADL007 | 15% | *Actinopolyspora xinjiangensis* DSM 46732 (jgi.1055186) | 99.71 | 13 |
| ADL008 | 15% | *Aidingimonas halophila* DSM 19219 (jgi.1107932) | 97.76 | 2 |
| ADL009 | 15% | *Aliifodinibius salicampi* KHM44 (LC198077) | 99.58 | 5 |
| ADL011 | 15% | *Alteribacillus alkaliphilus* JC229 (HG799487) | 98.12 | 1 |
| ADL012 | 5% | *Alteribacillus bidgolensis* IBRC-M10614 (jgi.1071278) | 99.86 | 7 |
| ADL013 | 15% | *Alteribacillus persepolensis* HS136 (FM244839) | 94.71 | 1 |
| ADL014 | 5% | *Anaerobacillus uncultured bacterium* F01CH1-61-65 (HF558583) | 96.51 | 1 |
| ADL015 | 15% | *Aquibacillus albus* YIM 93624 (JQ680032) | 100 | 9 |
| ADL017 | 5% | *Aquibacillus koreensis* BH30097 (AY616012) | 97.28 | 1 |
| ADL018 | 15% | *Aquisalimonas halophila* YIM 95345 (KC577145) | 100 | 12 |
| ADL019 | 15% | *Bacillus halmapalus* DSM 8723 (KV917375) | 99.29 | 6 |
| ADL020 | 15% | *Bacillus salarius* BH169 (AY667494) | 98.98 | 3 |
| ADL021 | 5% | *Bacillus swezeyi* NRRL B-41294 (MRBK01000096) | 99.79 | 6 |
| ADL022 | 15% | *Filobacillus milosensis* DSM 13259 (AJ238042) | 99.53 | 4 |
| ADL023 | 15% | *Caldalkalibacillus uzonensis* JW/WZ-YB58 (DQ221694) | 91.14 | 1 |
| ADL024 | 5% | *Glycomyces xiaoerkulensis* TRM 41368 (MF669725) | 99.72 | 2 |
| ADL026 | 15% | *Gracilibacillus bigeumensis* BH097 (EF520006) | 99.57 | 16 |
| ADL027 | 5%/15% | *Gracilibacillus orientalis* XH-63 (AM040716) | 98.36 | 2 |
| ADL028 | 15% | *Gracilibacillus saliphilus* YIM 91119 (EU784646) | 99.58 | 8 |
| ADL029 | 5%/15% | *Gracilibacillus thailandensis* TP2-8 (FJ182214) | 97.86 | 6 |
| ADL030 | 15% | *Gracilibacillus ureilyticus* MF38 (EU709020) | 97.95 | 1 |
| ADL031 | 15% | *Haloactinospora alba* YIM 90648 (DQ923130) | 99.44 | 3 |
| ADL032 | 15% | *Halobacillus dabanensis* D-8 (AY351395) | 99.11 | 5 |
| ADL033 | 5% | *Halobacillus yeomjeoni* MSS-402 (AY881246) | 98.52 | 3 |
| ADL034 | 15% | *Haloechinothrix halophila* YIM 93223 (KI632509) | 97.93 | 2 |
| ADL036 | 5% | *Halomonas arcis* AJ282 (EF144147) | 99 | 17 |
| ADL037 | 15% | *Halomonas lutea* DSM 23508 (ARKK01000003) | 100 | 4 |
| ADL038 | 5%/15% | *Halomonas xinjiangensis* TRM 0175 (JPZL01000008) | 99.5 | 21 |
| ADL040 | 5%/15% | *Jeotgalibacillus terrae* JSM 081008 (FJ527421) | 99.15 | 4 |
| ADL041 | 5% | *Kocuria assamensis* S9-65 (HQ018931) | 99.64 | 1 |
| ADL042 | 5% | *Kocuria palustris* DSM 11925 (Y16263) | 99.72 | 1 |
| ADL044 | 5% | *Longimycelium tulufanense* TRM 46004 (HQ229000) | 100 | 5 |
| ADL045 | 5% | *Marinactinospora thermotolerans* DSM 45154 (FUWS01000037) | 99.29 | 6 |
| ADL047 | 5% | *Marinobacter guineae* M3B (AM503093) | 98.59 | 3 |
| ADL048 | 5%/15% | *Marinobacter lacisalsi* FP2.5 (EU047505) | 98.9 | 5 |
| ADL049 | 15% | *Marinococcus luteus* DSM 23126 (jgi.1089306) | 100 | 11 |
| ADL050 | 5% | *Micromonospora andamanensis* SP03-05 (JX524154) | 99.05 | 2 |
| ADL053 | 5% | *Micromonospora halotolerans* CR18(NR_132303.1) | 100 | 3 |
| ADL054 | 5% | *Myceligenerans salitolerans* XHU 5031 (JX316007) | 100 | 2 |
| ADL055 | 15% | *Nesterenkonia halophila* YIM 70179 (AY820953) | 99 | 2 |
| ADL056 | 5% | *Nitratireductor shengliensis* 110399 (KC222645) | 97.58 | 3 |
| ADL057 | 5% | *Nocardiopsis aegyptia* DSM 44442 (AJ539401) | 99.43 | 7 |
| ADL060 | 5% | *Nocardiopsis mwathae* No.156 (KF976731) | 98.72 | 3 |
| ADL061 | 15% | *Nocardiopsis rosea* YIM 90094 (AY619713) | 99.27 | 6 |

*(Continued)*

**Table 3.** (Continued)

| Isolate | Salinity | Closest cultivated species (GenBank accession no.) | Similarity (%) | No. of isolates |
|---|---|---|---|---|
| ADL063 | 5% | *Nocardiopsis sinuspersici* HM6 (EU410476) | 98.8 | 1 |
| ADL065 | 5% | *Ornithinibacillus scapharcae* TW25 (AEWH01000025) | 98.52 | 1 |
| ADL066 | 15% | *Phytoactinopolyspora halotolerans* YIM 96448 (KY979511) | 100 | 3 |
| ADL067 | 15% | *Piscibacillus halophilus* HS224 (FM864227) | 99.01 | 5 |
| ADL068 | 5% | *Planococcus salinarum* DSM 23820 (MBQG01000128) | 98.82 | 2 |
| ADL069 | 15% | *Pontibacillus marinus* BH030004 (AVPF01000156) | 99.12 | 9 |
| ADL070 | 5% | *Prauserella marina* CGMCC 4.5506 (jgi.1085010) | 97.3 | 1 |
| ADL071 | 15% | *Saccharomonospora azurea* NA-128 (AGIU02000033) | 100 | 6 |
| ADL073 | 15% | *Saccharomonospora xiaoerkulensis* TRM 41495 (KU511278) | 99.72 | 1 |
| ADL075 | 15% | *Saccharopolyspora lacisalsi* TRM 40133 (JF411070) | 100 | 5 |
| ADL076 | 15% | *Salinicoccus luteus* YIM 70202 (DQ352839) | 100 | 9 |
| ADL078 | 5% | *Salinifilum aidingensis* TRM 46074 (JX193858) | 99.93 | 4 |
| ADL079 | 5% | *Sediminibacillus halophilus* EN8d (AM905297) | 100 | 11 |
| ADL080 | 15% | *Sinobaca qinghaiensis* YIM 70212 (DQ168584) | 100 | 5 |
| ADL082 | 5% | *Streptomyces aidingensis* TRM46012 (HQ286045) | 100 | 6 |
| ADL083 | 5% | *Streptomyces ambofaciens* ATCC 23877 (CP012382) | 99.31 | 2 |
| ADL084 | 5% | *Streptomyces asenjonii* KNN 35.1b (LT621750) | 98.91 | 3 |
| ADL086 | 5% | *Streptomyces coelicoflavus* NBRC 15399 (AB184650) | 99.79 | 3 |
| ADL087 | 5% | *Streptomyces fukangensis* EGI 80050 (KF040416) | 98.6 | 2 |
| ADL088 | 5% | *Streptomyces griseoincarnatus* LMG 19316 (AJ781321) | 99.93 | 7 |
| ADL090 | 5% | *Streptomyces xinghaiensis* S187 (CP023202) | 99.93 | 4 |
| ADL091 | 5%/15% | *Virgibacillus sediminis* YIM kkny3 (AY121430) | 99.65 | 11 |
| ADL092 | 5% | *Zhihengliuella somnathii* JG 03 (EU937748) | 99.17 | 2 |
| XHU5135 | 15% | *Aidingimonas halophila* BH017 (EU191906) | 97.52 | 1 |

two strains had such low degrees of sequence similarity, suggesting that these may represent two novel genera of *Bacillaceae*. In the study, *Halomonas* (42 strains, 12.2%), *Gracilibacillus* (33 strains, 9.6%), *Streptomyces* (27 strains, 7.9%), *Actinopolyspora* (26 strains, 7.6%), *Nocardiopsis* (17 strains, 5.0%), *Bacillus* (12 strains, 4.4%), *Aquisalimonas* (12 strains, 3.5%), *Marinococcus* (11 strains, 3.2%), *Virgibacillus* (11 strains, 3.2%), and *Sediminibacillus* (11 strains, 3.2%) were some dominant group in sediment samples of Aiding Lake. The number of microorganisms in each of the other genus is relatively small (Table 3; Fig 3). For example, *Anaerobacillus*, *Prauserella*, and *Ornithinibacillus* include only one strain, respectively. To isolate halophilic or halotolerant bacteria in the sediments of Aiding Lake, the agar plates were supplemented with 5% or 15% (w/v) NaCl. Approximately 141 strains isolated from these media with 5% NaCl belonged to 25 different genera, and 202 strains isolated from the media using 15% NaCl belonged to 23 different genera (Table 3).

## 3.3 Bacterial isolates from different media

To obtain additional bacterial groups, the sediment samples from Aiding Lake were isolated using nine different media (Table 1). Five class of bacteria, namely, *Actinobacteria*, *Bacilli*, *Alphaproteobacteria*, *Gammaproteobacteria*, and *Balneolia*, including 14 orders, 17 families, and 43 genera (including 2 novel genera) were obtained (Table 4). Most of the bacterial groups were isolated using microcrystalline cellulose-sorbitol agar (G), and 13 bacterial genera (*Aquibacillus*, *Aquisalimonas*, *Bacillus*, *Filobacillus*, *Gracilibacillus*, *Halomonas*, *Kocuria*, *Nocardiopsis*, *Ornithinibacillus*, *Phytoactinopolyspora*, *Piscibacillus*, *Virgibacillus*, and *Zhihengliuella*)

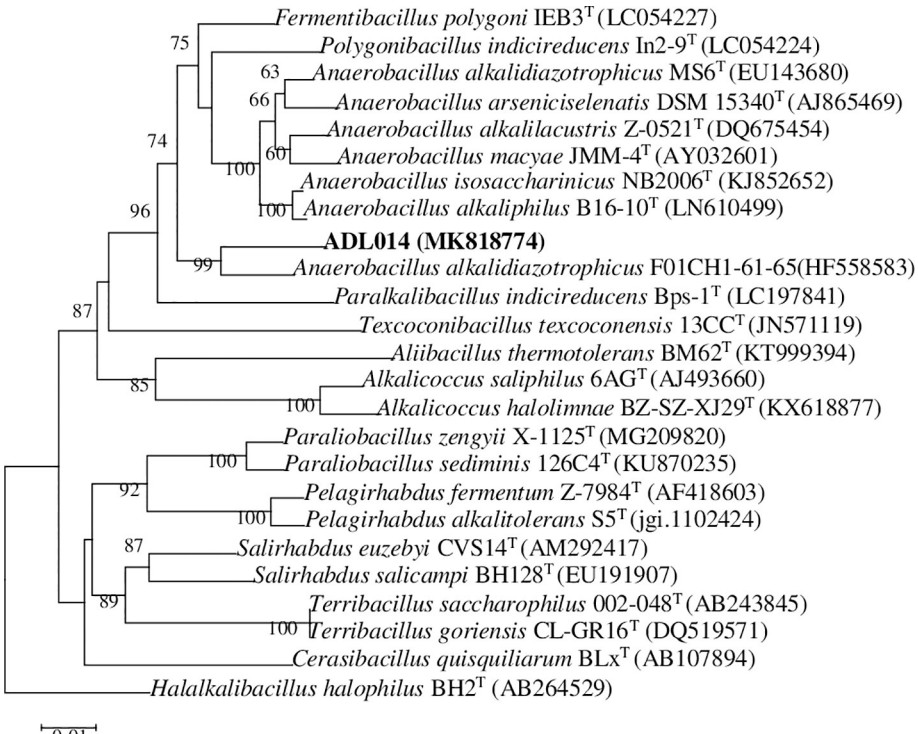

**Fig 1. Phylogenetic tree of strain ADL014 and its near neighbors calculated from 16S rRNA gene sequences using Kimura's evolutionary distance method (Kimura, 1980) and the neighbor-joining method of Saitou and Nei (1987).** Bar, 0.01 nucleotide substitutions per site.

were isolated. At the same time, microcrystalline cellulose-proline agar (C), stachyose tetrahydrate-alanine agar (F), casein hydrolysate acid-starch agar (B), and glycerin-asparagine agar (D) resulted in relatively efficient isolations for 11 genera (*Actinopolyspora*, *Anaerobacillus*, *Glycomyces*, *Haloactinospora*, *Jeotgalibacillus*, *Kocuria*, *Myceligenerans*, *Nesterenkonia*, *Pontibacillus*, *Saccharomonospora*, and *Salinifilum*), 8 genera (*Aquibacillus*, *Halobacillus*, *Haloechinothrix*, *Halomonas*, *Marinactinospora*, *Marinobacter*, *Nocardiopsis*, and *Streptomyces*), 7 genera (*Actinopolyspora*, *Halobacillus*, *Marinobacter*, *Micromonospora*, *Planococcus*, *Saccharopolyspora*, and *Streptomyces*), and 6 genera (*Actinopolyspora*, *Longimycelium*, *Marinococcus*, *Nocardiopsis*, *Saccharomonospora*, and *Streptomyces*), respectively. Only two known genera (*Gracilibacillus* and *Sinobaca*) and one novel genus (ADL013) were isolated using yeast extract-glycerin agar (I). The results of isolation using the other media are shown in Table 4. From the number of isolated strains, 95, 57, and 57 strains were isolated using media G, F, and C, respectively, and the number of strains isolated from media A, B, D, E, H and I was relatively small (Tables 3 and 4). In addition, the bacterial diversity recovered using different media also varied considerably. Medium G had the best recoverability, with 14 of the total bacterial genera recovered. Media I and E showed the lowest recoverability at the genus level (Table 4). In short, to obtain more bacterial resources, it is also necessary to develop different types of media.

## 3.4 Bacterial isolates from different sediments and Pearson correlation

The number of isolates of bacteria recovered from three sediments (S1, S2, and S3) are different.164, 88 and 91 strains were isolated from sediment samples S1, S2, and S3, respectively (S1

**Table 4. Statistical analyses of the relationships between the taxa of the bacterial strains and the nine different media.**

| Medium | No. of isolates | Isolate taxon | | | |
|---|---|---|---|---|---|
| | | Phylum | Order | Family | Genus |
| A | 27 | Actinobacteria | Micromonosporales | Micromonosporaceae | Micromonospora |
| | | | Pseudonocardiales | Pseudonocardiaceae | Prauserella |
| | | | Streptomycetales | Streptomycetaceae | Streptomyces |
| | | | Streptosporangiales | Nocardiopsaceae | Nocardiopsis |
| B | 24 | Actinobacteria | Actinopolysporales | Actinopolysporaceae | Actinopolyspora |
| | | | Micromonosporales | Micromonosporaceae | Micromonospora |
| | | | Pseudonocardiales | Pseudonocardiaceae | Saccharopolyspora |
| | | | Streptomycetales | Streptomycetaceae | Streptomyces |
| | | Firmicutes | Bacillales | Bacillaceae | Halobacillus |
| | | Proteobacteria | Alteromonadales | Planococcaceae | Planococcus |
| | | | | Alteromonadaceae | Marinobacter |
| C | 57 | Actinobacteria | Actinopolysporales | Actinopolysporaceae | Actinopolyspora |
| | | | Glycomycetales | Glycomycetaceae | Glycomyces |
| | | | Micrococcales | Micrococcaceae | Kocuria |
| | | | | | Nesterenkonia |
| | | | Pseudonocardiales | Promicromonosporaceae | Myceligenerans |
| | | | Streptosporangiales | Pseudonocardiaceae | Saccharomonospora |
| | | | | Nocardiopsaceae | Haloactinospora |
| | | Firmicutes | Bacillales | Bacillaceae | Salinifilum |
| | | | | | Anaerobacillus |
| | | | | | Jeotgalibacillus |
| | | | | | Pontibacillus |
| D | 25 | Actinobacteria | Actinopolysporales | Actinopolysporaceae | Actinopolyspora |
| | | | Pseudonocardiales | Pseudonocardiaceae | Longimycelium |
| | | | Streptomycetales | Streptomycetaceae | Streptomyces |
| | | | Streptosporangiales | Nocardiopsaceae | Saccharomonospora |
| | | | | | Nocardiopsis |
| | | Firmicutes | Bacillales | Bacillaceae | Marinococcus |
| E | 20 | Rhodothermaeota | Balneolales | Balneolaceae | Aliifodinibius |
| | | Firmicutes | Bacillales | Bacillaceae | Bacillus |
| | | | | Staphylococcaceae | Salinicoccus |
| F | 57 | Actinobacteria | Pseudonocardiales | Pseudonocardiaceae | Haloechinothrix |
| | | | Streptomycetales | Streptomycetaceae | Streptomyces |
| | | | Streptosporangiales | Nocardiopsaceae | Marinactinospora |
| | | | | | Nocardiopsis |
| | | Firmicutes | Bacillales | Bacillaceae | Aquibacillus |
| | | | | | Halobacillus |
| | | Proteobacteria | Alteromonadales | Marinobacter family | Marinobacter |
| | | | Oceanospirillales | Halomonadaceae | Halomonas |
| G | 95 | Actinobacteria | Jiangellales | Jiangellaceae | Phytoactinopolyspora |
| | | | Micrococcales | Micrococcaceae | Kocuria |
| | | | | | Zhihengliuella |
| | | | Streptosporangiales | Nocardiopsaceae | Nocardiopsis |
| | | Firmicutes | Bacillales | Bacillaceae | Bacillus |
| | | | | | Aquibacillus |
| | | | | | Filobacillus |

*(Continued)*

**Table 4.** (Continued)

| Medium | No. of isolates | Isolate taxon | | | |
|---|---|---|---|---|---|
| | | Phylum | Order | Family | Genus |
| | | | | | *Gracilibacillus* |
| | | | | | *Ornithinibacillus* |
| | | | | | *Piscibacillus* |
| | | | | | *Virgibacillus* |
| | | *Proteobacteria* | *Chromatiales* | *Nitrococcus family* | *Aquisalimonas* |
| | | | *Oceanospirillales* | *Halomonadaceae* | *Halomonas* |
| H | 24 | *Firmicutes* | *Bacillales* | *Bacillaceae* | *Alteribacillus* |
| | | | | | *Gracilibacillus* |
| | | | | | ADL023 |
| | | *Proteobacteria* | *Rhizobiales* | *Phyllobacteriaceae* | *Nitratireductor* |
| | | | *Oceanospirillales* | *Halomonadaceae* | *Aidingimonas* |
| I | 14 | *Firmicutes* | *Bacillales* | *Bacillaceae* | *Gracilibacillus* |
| | | | | | *Sinobaca* |
| | | | | | ADL013 |

Table). In sediment sample S1, 28 bacterial genera were isolated, while another 27, and 24 bacterial genera were isolated from samples S2 and S3, respectively (S1 Table). Although there was a small difference in the number of genera per sample, there was a large difference in the type of genera per sample (S1 Table). For example, *Haloactinospora*, *Nesterenkonia*, *Phytoactinopolyspora*, *Saccharomonospora*, *Saccharopolyspora*, ADL013, and ADL023 were isolated only from sample S1; *Glycomyces*, *Haloechinothrix*, *Myceligenerans*, *Ornithinibacillus*, and *Prauserella* were isolated only from sample S2; meanwhile, *Anaerobacillus*, *Kocuria*, *Salinifilum*, and *Zhihengliuella* were also isolated only from sample S3 (S1 Table). In addition, *Actinopolyspora*, *Aquibacillus*, *Aquisalimonas*, *Bacillus*, *Gracilibacillus*, *Halomonas*, *Marinococcus*, *Nocardiopsis*, *Salinicoccus*, *Streptomyces*, and *Virgibacillus* were distributed in all three samples (S1 Table). The results showed that the number and diversity of bacteria were different even in different sites from the same salt lake.

Pearson's correlation analysis revealed that $Na^+$, $K^+$, $Cl^-$ and $HCO_3^-$ were mainly positively correlated with the relative abundances of *Actinopolyspora*, *Gracilibacillus*, *Pontibacillus*, *Aidingimonas*, *Aquisalimonas*, *Halobacillus*, *Bacillus*, *Haloactinospora*, *Nesterenkonia*, *Phytoactinopolyspora*, *Saccharomonospora*, *Saccharopolyspora*, ADL013, ADL023, *Alteribacillus*, and *Nocardiopsis*, respectively (Table 5). $Mn^{2+}$ was mainly positively correlated with the relative abundances of *Aquibacillus*, *Filobacillus*, and *Marinococcus* but negatively correlated with *Micromonospora*. $Fe^{2+}$ was mainly positively correlated with the relative abundances of *Marinactinospora*. $SO_4^{2-}$ was mainly positively correlated with the relative abundances of *Salinicoccus* but negatively correlated with *Piscibacillus* (Table 5).

## 4 Discussion

Bacterial diversity in Aiding Lake was based on the 16S rRNA gene sequences, which was relatively higher than other salt lakes at the genus level. For example, sequencing of 16S rRNA genes indicated the presence of members of bacterial genera *Bacillus*, *Halomonas*, *Pseudomonas*, *Exiguobacterium*, *Vibrio*, *Paenibacillus*, and *Planococcus* in the salt lake La Sal del Rey, in extreme South Texas (USA) [33]. Previous studies also have shown that bacterial diversity in other saline lake ecosystems were mainly composed of the bacteria (including 16 genera:

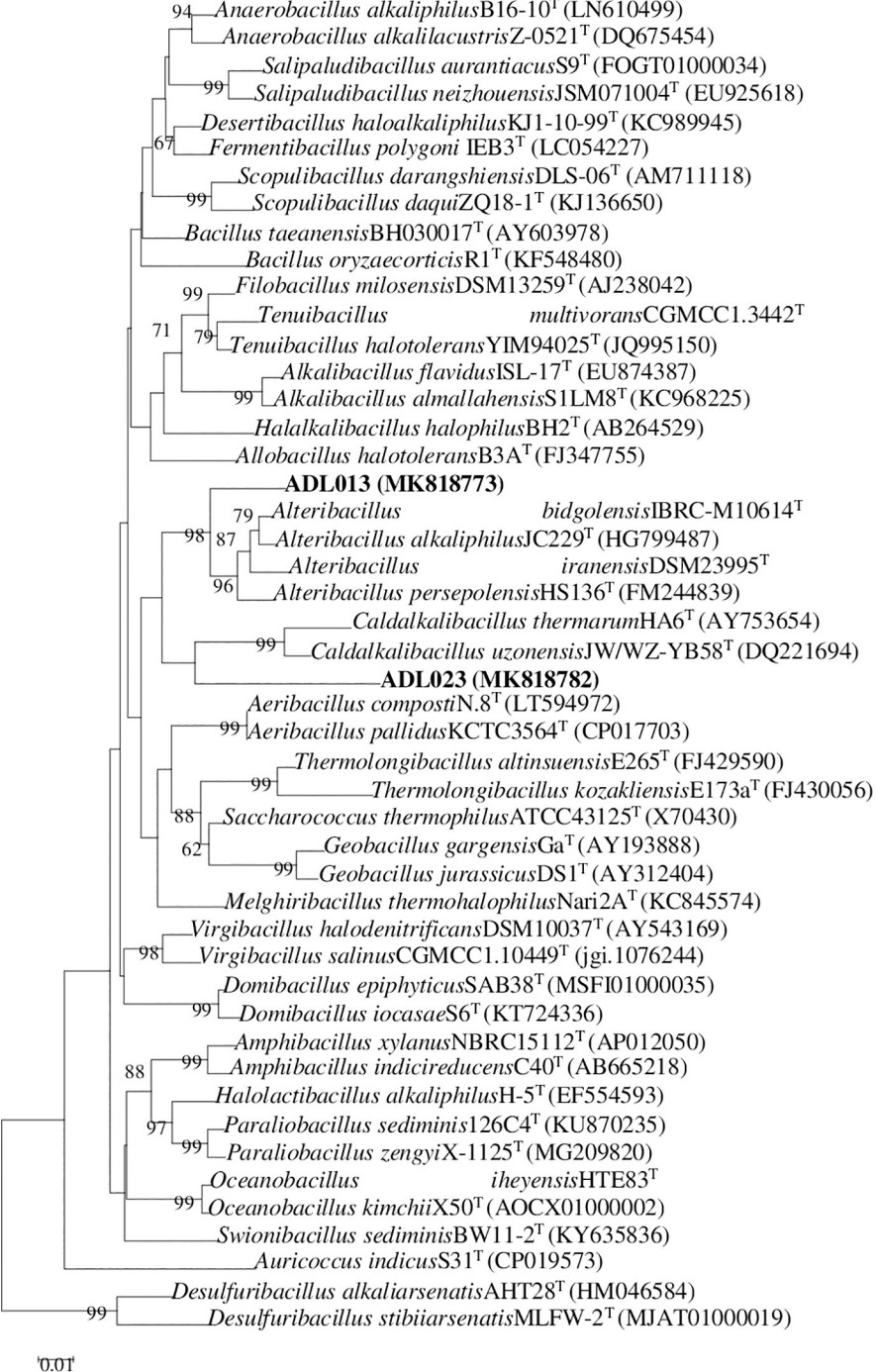

**Fig 2. Phylogenetic dendrogram for taxa of the family *Bacillaceae* reconstructed using the neighbor-joining method based on almost complete 16S rRNA gene sequences to display the taxonomic position of strain ADL013 or strain ADL023.** Numbers at nodes indicate levels of bootstrap support (%) based on neighbor-joining analysis of 1000 resampled datasets; only values above 50% are shown. Bar, 0.01 nucleotide substitutions per site.

*Bacillus, Chromohalobacter, Gracilibacillus, Halobacillus, Halolactibacillus, Halomonas, Halovibrio, Idiomarina, Oceanobacillus, Piscibacillus, Salicola, Salimicrobium, Salinicoccus, Staphylococcus, Thalassobacillus,* and *Virgibacillus*) [34, 35]. Notably, some rare genera from the

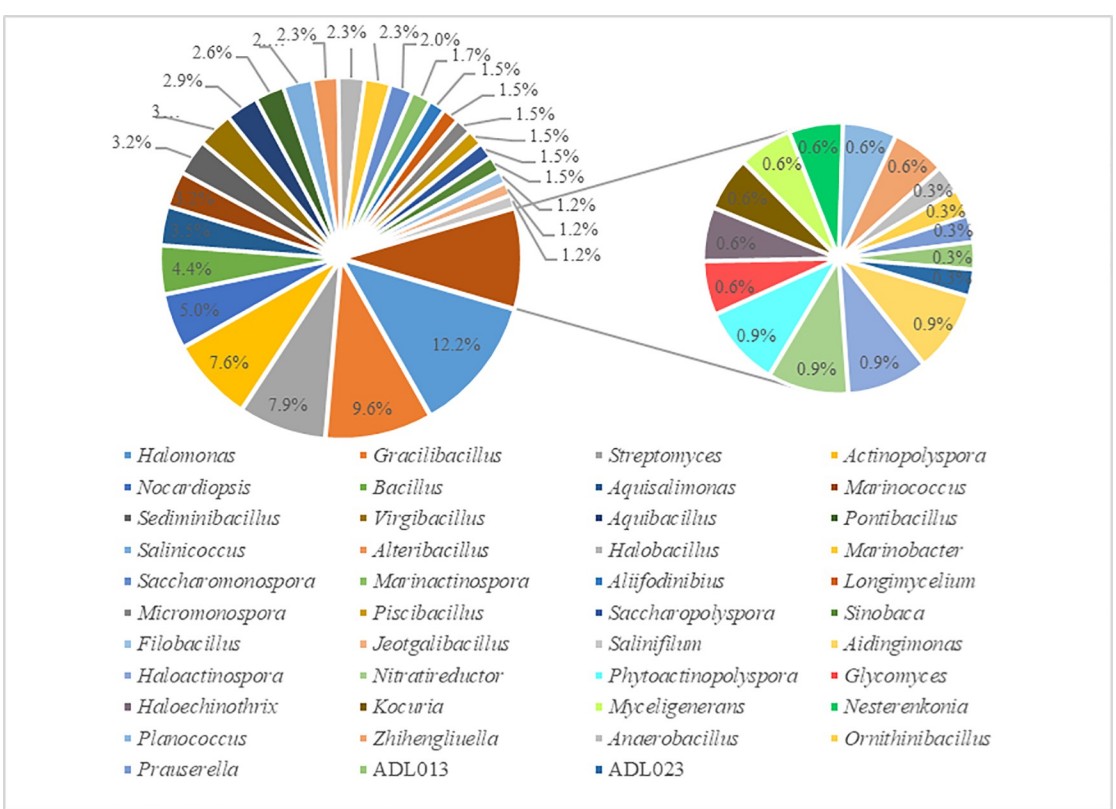

**Fig 3. Percentage of isolated strains in each genus.**

sediments of Aiding Lake such as *Aidingimonas*, *Aliifodinibius*, *Filobacillus*, *Haloechinothrix*, *Jeotgalibacillus*, *Longimycelium*, *Myceligenerans*, *Ornithinibacillus*, *Phytoactinopolyspora*, and *Piscibacillus* were discovered in the present study. In addition, phylum *Rhodothermaeota* was detected for the first time in sediment samples from a salt lake.

The richness and diversity of actinobacteria isolated from Aiding Lake in the present study were relatively high, consisting of eight orders (*Actinopolysporales*, *Glycomycetales*, *Jiangellales*, *Micrococcales*, *Micromonosporales*, *Pseudonocardiales*, *Streptomycetales* and *Streptosporangiales*). Wu et al. (2008) have found six actinobacterial orders using pure culture in salt lake, hypersaline spring, and salt mine of China, and these orders were *Actinopolysporales*, *Glycomycetales*, *Micrococcales*, *Pseudonocardiales*, *Streptomycetales*, and *Streptosporangiales* [36]. Our results suggested that the diversity of actinobacterial 16S rRNA gene sequences of strains from Aiding Lake were more diverse at the genus level than those reported in saline environment. For examples, twelve genera, namely *Actinomadura*, *Actinomycetospora*, *Microbispora*, *Micromonospora*, *Mycobaterium*, *Nocardia*, *Nocardiopsis*, *Pseudonocardia*, *Saccharopolyspora*, *Sphaerosporangium*, *Streptomyces*, and *Streptosporangium* were isolated from marine sponges in Florida, USA [37], and ten genera, including *Actinotalea*, *Arthrobacter*, *Brachybacterium*, *Brevibacterium*, *Kocuria*, *Kytococcus*, *Microbacterium*, *Micrococcus*, *Mycobacterium*, and *Pseudonocardia* from Arctic marine sediments were obtained by culture-dependent approaches [38]. Atika et al. isolated fifty-two halophilic actinomycetes belonging to the *Actinopolyspora*, *Nocardiopsis*, *Saccharomonospora*, *Saccharopolyspora*, and *Streptomonospora* genera from Saharan saline soils of Algeria [39]. Ronoh et al. identified four genera (*Dietzia*, *Microbacterium*, *Nocardia*, and *Rhodococcus*) of actinobacteria from Lake Magadi, Kenya [40]. However,

**Table 5. Pearson correlation coefficient (*r*) and *p*-value for isolated bacterial genera.** Only correlations with $p \leq 0.05$ are shown.

| Genus | Correlation with | *p*-Value | *r* |
|---|---|---|---|
| *Actinopolyspora* | $Na^+$ | 0.04 | 0.99 |
| *Aidingimonas* | $K^+$ | 0.02 | 0.99 |
| *Aliifodinibius* | $SO_4^{2-}$ | 0.02 | 0.99 |
| *Alteribacillus* | $HCO_3^-$ | 0.04 | 0.99 |
| *Aquibacillus* | $Mn^{2+}$ | 0.00 | 1.00 |
| *Aquisalimonas* | $K^+$ | 0.02 | 0.99 |
| *Bacillus* | $Cl^-$ | 0.02 | 1.00 |
| *Filobacillus* | $Mn^{2+}$ | 0.00 | 1.00 |
| *Gracilibacillus* | $Na^+$ | 0.04 | 0.99 |
| *Haloactinospora* | $Cl^-$ | 0.02 | 1.00 |
| *Halobacillus* | $K^+$ | 0.05 | 0.99 |
| *Marinactinospora* | $Fe^{2+}$ | 0.01 | −1.00 |
| *Marinococcus* | $Mn^{2+}$ | 0.00 | 1.00 |
| *Micromonospora* | $Mn^{2+}$ | 0.00 | −1.00 |
| *Nesterenkonia* | $Cl^-$ | 0.02 | 1.00 |
| *Nocardiopsis* | pH, $HCO_3^-$ | 0.04, 0.02 | −0.99, 1.00 |
| *Phytoactinopolyspora* | $Cl^-$ | 0.02 | 1.00 |
| *Piscibacillus* | $SO_4^{2-}$ | 0.02 | −0.99 |
| *Pontibacillus* | $Na^+$, $Cl^-$, | 0.05, 0.05 | 0.99, 0.99 |
| *Saccharomonospora* | $Cl^-$ | 0.02 | 1.00 |
| *Saccharopolyspora* | $Cl^-$ | 0.02 | 1.00 |
| *Salinicoccus* | $SO_4^{2-}$ | 0.02 | 0.99 |
| ADL013 | $Cl^-$ | 0.02 | 1.00 |
| ADL023 | $Cl^-$ | 0.02 | 1.00 |

18 genera, namely *Actinopolyspora*, *Glycomyces*, *Haloactinospora*, *Haloechinothrix*, *Kocuria*, *Longimycelium*, *Marinactinospora*, *Micromonospora*, *Myceligenerans*, *Nesterenkonia*, *Nocardiopsis*, *Phytoactinopolyspora*, *Prauserella*, *Saccharomonospora*, *Saccharopolyspora*, *Salinifilum*, *Streptomyces*, and *Zhihengliuella* were isolated in this study. Therefore, bacterial diversity may differ between different sample collection regions and different saline samples. Of course, it is also possible that the diversity of isolated bacteria may vary due to different media or isolated technology.

In addition to strains ADL013, ADL014, and ADL023 are novel species because of low similarity, the 16S rRNA gene sequences of strain XHU5135 showed 97.52% identities to the nearest neighbors, *Aidingimonas halophila* YIM 90637 [41]. Although it showed higher 16S rRNA gene similarities (97.52%) to the closest recognized strains, DNA-DNA hybridization experiments revealed that levels of DNA-DNA relatedness between strain XHU 5135 and *Aidingimonas halophila* YIM 90637 were 23.3± 2.8%. This values is below the 70% cut-off point recommended for recognition of genomic species [42]. Thus, strain XHU 5135 represents a novel species of the genus *Aidingimonas*. The neighbour-joining phylogenetic tree based on 16S rRNA gene sequences of strain XHU 5135 and other related species is shown in Fig 4 with high levels of bootstrap support. At present, we isolated and cultured 71 species (including 4 novel genera) of 43 bacterial genera (including 2 novel genera). To our knowledge, this is the first study to recover such a high diversity of culturable bacteria from salt lake sediments. In addition, *Aidingimonas halophila* YIM 90637 [41], *Streptomyces aidingensis* TRM46012 [43], *Longimycelium tulufanense* TRM 46004 [44], *Salinifilum aidingensis* TRM 46074 [45], and

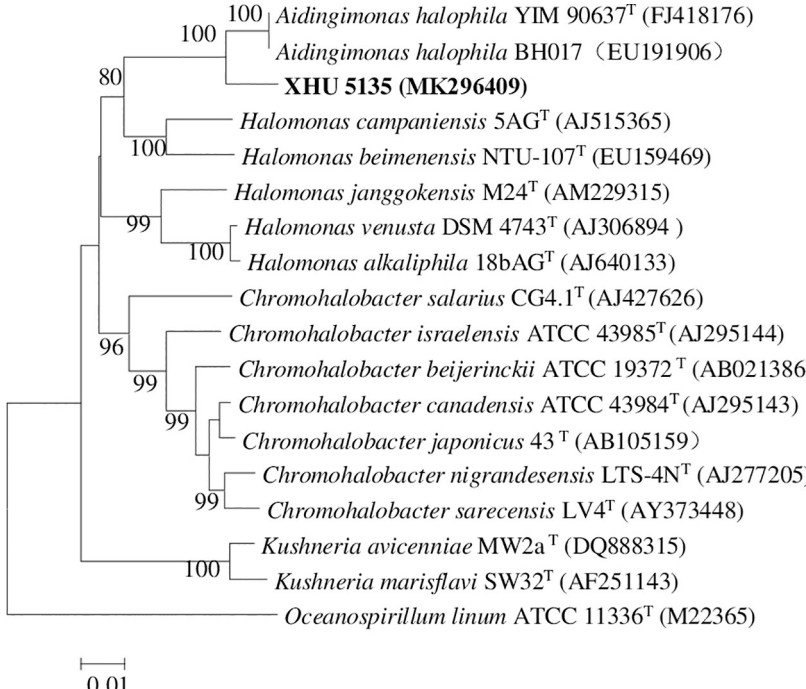

**Fig 4. Neighbour-joining tree based on 16S rRNA gene sequences, showing the phylogenetic relationships of the novel isolate XHU 5135 and related taxa.** Numbers at nodes are bootstrap percentages based on 1000 replicates. Bar, 0.01 nucleotide substitutions per site.

*Gracilibacillus aidingensis* YIM 98001 [46] were also isolated from Aiding Lake, and above strains have been characterized as some novel species. These results indicate that there are potentially unique, novel sources of bacteria in Aiding Lake.

The dry lake has a high NaCl content. However, a large number of bacteria are isolated using the media employed a lower content of NaCl (5%). Possible reasons for such inconsistency could be twofold: (1) bacteria may tolerate a large range of salinity. For example, *Halomonas xinjiangensis* TRM 0175 could grow at 0–20% NaCl [47]; (2) surface layer of the soil samples have a high salinity, while the other parts have a low salinity.

Generally, bacterial diversity in extreme environments is relatively low. Therefore, according to the differences in the utilization of carbon and nitrogen sources by microorganisms, as well as the high repetition rate of microorganisms isolated from conventional media, we designed some media containing rare carbon and nitrogen source (microcrystalline cellulose, glycerin, stachyose tetrahydrate, sorbitol, Beta-Cyclodextrin, casein hydrolysate acid, proline, arginine, asparagine, or alanine) in the hope of mining more novel species and presenting better bacterial diversity. To our satisfaction, an unexpectedly high bacterial diversity was observed from sediment samples from Aiding Lake. Forty-three bacterial genera were identified. The results indicate that it is feasible to use rare carbon and nitrogen source media to mine more species, and some strains might represent a valuable source of new species, thereby providing a new reference for further understanding bacterial diversity in hypersaline environments. This study also suggested that the diversity of bacteria isolated from Aiding Lake is largely dependent on the isolation media. Obviously, the number and diversity of bacteria isolated from different media are different in this study. Although media I and E showed the lowest recoverability at the genus level, the microorganism of *Sinobaca* genus was only isolated using medium I, and a novel strain ADL013 was found using the medium too. At the same

time, the microorganism of phylum *Rhodothermaeota* was detected only using medium E. Of all the media, the most actinobacterial genera (7) were isolated from medium C. This result suggests that medium C is suitable for the isolation of actinomycetes in sediments of salt lake. In addition, the most abundant bacterial genera were obtained from medium G (Microcrystalline cellulose-sorbitol agar), this may indicate that these bacteria have a special utilization for microcrystalline cellulose, or sorbitol, or Beta-Cyclodextrin, but at present the mechanism of why medium G is suitable for the isolation of bacteria from Aiding lake is unclear. No universal medium or uniform isolation technology has been established for microbial resources around the world. Therefore, there is a need to develop novel isolation media or isolation techniques to better mine non-culturable bacterial resources.

## Supporting information

**S1 Fig. Geographic location of Aiding Lake.**
(TIF)

**S1 Table. The number of isolates recovered from each sediment.**
(DOCX)

## Author Contributions

**Data curation:** Tong-Wei Guan, Yi-Jin Lin, Ke-Bao Chen.

**Formal analysis:** Meng-Ying Ou.

**Writing – original draft:** Tong-Wei Guan.

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
