## [Decision Letter · Decision Letter 0]

23 Dec 2019

PONE-D-19-20101

Isolation and Diversity of Sediment Bacteria in the Hypersaline Aiding Lake,
China

PLOS ONE

Dear Dr Guan,

Thank you for submitting your manuscript to PLOS ONE. After careful consideration, we
feel that it has merit but does not fully meet PLOS ONE’s publication criteria as it
currently stands. Therefore, we invite you to submit a revised version of the
manuscript that addresses the points raised during the review process.

There are several critical points raised by the two referees and I would appreciate
if you comply their comments/suggestions when resubmitting your manuscript which,
again, I will be very happy to follow.

The full referee’s comments are indicated at the end of this letter.

We would appreciate receiving your revised manuscript by Feb 06 2020 11:59PM. When
you are ready to submit your revision, log on to https://www.editorialmanager.com/pone/ and select the 'Submissions
Needing Revision' folder to locate your manuscript file.

If you would like to make changes to your financial disclosure, please include your
updated statement in your cover letter.

To enhance the reproducibility of your results, we recommend that if applicable you
deposit your laboratory protocols in protocols.io, where a protocol can be assigned
its own identifier (DOI) such that it can be cited independently in the future. For
instructions see: http://journals.plos.org/plosone/s/submission-guidelines#loc-laboratory-protocols

We look forward to receiving your revised manuscript.

Kind regards,

Luis Angel Maldonado Manjarrez, Ph.D.

Academic Editor

PLOS ONE

Journal Requirements:

3. We note that you are reporting an analysis of a microarray, next-generation
sequencing, or deep sequencing data set. PLOS requires that authors comply with
field-specific standards for preparation, recording, and deposition of data in
repositories appropriate to their field. Please upload these data to a stable,
public repository (such as ArrayExpress, Gene Expression Omnibus (GEO), DNA Data
Bank of Japan (DDBJ), NCBI GenBank, NCBI Sequence Read Archive, or EMBL Nucleotide
Sequence Database (ENA)). In your revised cover letter, please provide the relevant
accession numbers that may be used to access these data. For a full list of
recommended repositories, see http://journals.plos.org/plosone/s/data-availability#loc-omics or http://journals.plos.org/plosone/s/data-availability#loc-sequencing.

Reviewers' comments:

Reviewer's Responses to Questions

**Comments to the Author**

1. Is the manuscript technically sound, and do the data support the conclusions?

Reviewer #1: Partly

Reviewer #2: Partly

2. Has the statistical analysis been performed
appropriately and rigorously? 

Reviewer #1: N/A

Reviewer #2: N/A

3. Have the authors made all data underlying the
findings in their manuscript fully available?

Reviewer #1: Yes

Reviewer #2: No

4. Is the manuscript presented in an intelligible
fashion and written in standard English?

Reviewer #1: Yes

Reviewer #2: No

5. Review Comments to the Author

Reviewer #1: This manuscript describes the isolation of bacteria from sediments
collected from a hypersaline lake. A diversity analysis of these bacterial isolates
based on the evolutionary relatedness of the 16S rRNA gene was also performed.
Overall, the study presents new and interesting data with regards to the potential
to access novel species and genera via a culture-based approach and highlights the
value of exploring environments that are unique and underexplored. Here are some
comments and recommendations for the authors:

1. Abstract: It would be helpful to the reader to understand the aim of the study at
the outset and therefore should be included in the Abstract.

2. Line 30: Insert the word ‘supplemented’ before ‘with 5% of 15% (w/v) NaCl’

3. Line 31: The use of the word ‘significantly’ implies statistical analyses – if any
were performed, please provide details in the methodology section and provide the
p-value.

4. Line 37: Values below 20 is typically written out in full, therefore it should be
‘…two novel genera…’ and ‘…four novel species…’

5. Line 38: Since reference is made to more than one genus, the plural, ‘genera’,
should be used.

6. Line 53: The sentence is ambiguous, implying that the novel species described are
different forms of culture-dependent methods. To avoid the ambiguity, it is
recommended that the sentence should be changed to: ‘…have been described using
culture-dependent methods: Brevibacterium salitolerans…’

7. Line 59: It is mentioned that the Turpan Basin is the hottest region in China –
can you provide a temperature range to provide insight into the environment from
which the samples were collected?

8. Line 60: Please provide a reference to support the statements regarding
temperature and salinity.

9. Line 62: ‘bacteria diversity’ should be ‘bacterial diversity’

10. Site description and sample collection: Are three sediment samples sufficient to
serve as representative samples of the whole lake area? If they were not meant to be
representative samples, then I would suggest changing the manuscript title to
reflect this.

11. Line 77: Sediments will not dissolve in distilled water – rather state that the
sediments ‘were resuspended in distilled water’.

12. Line 86: What was the rationale behind the design of the different types of
isolation media? Even though emphasis is made that these different types of media
allowed for access to unique bacterial isolates, the significance of this is not
really discussed or elaborated on in the manuscript. The section on this in the
Discussion should therefore be expanded on. In addition, why was the particular
pre-treatment method employed in this study? Do the authors feel that this also
contributed towards the isolation of a wider variety of bacterial strain?

13. Lines 87-88: Stating that ‘The compositions of the nine media are shown in Table
1’ is redundant and should be deleted. Reference to the media have already been made
in line 86.

14. Line 90: Is there any particular reason why an incubation temperature of
37�C was used? It is 12-14 degrees higher than the temperature recorded
for the areas where the sediment samples were collected.

15. Line 111: The use of a 99% sequence identity based on 16S rRNA gene sequences as
the basis for clustering into one operational taxonomic unit poses great risk of
clustering isolates that are novel. It is often found in actinobacterial genera in
particular where unique species share a high 16S rRNA sequence similarity, sometimes
even 100%. An alternative method of dereplication should’ve been employed – either a
sequence-based method (use of another taxonomic marker) or culture-based (phenotypic
differences).

16. Lines 119-125: Does the Geology of the region support the differences in the
physicochemical properties of the three sediment samples? Is there any correlation
between the isolates (genus/family) and the sediment sample isolated from?

17. Lines 145-148: Make sure that all mention of ‘strains’ are correct – in some
instances, the singular, ‘strain’ is used instead of the plural form.

18. Lines 155-156: Correct the sentence structure – ‘…suggesting that these may
represent two novel genera of Bacillaceae’.

19. Line 170: ’17 families’ not ’17 family’.

20. Line 171-172: A statement is made here that most bacterial groups were isolated
on one specific medium – this needs to be expanded on in the discussion section.

21. Line 184: ‘…one novel genera’ should read ‘…one novel genus’.

22. Line 195: ‘…16S rDNA…’ should be ‘…16S rRNA genes…’

23. In the discussion section, mention is made of the various genera isolated from
other types of salt lakes. However, different media and isolation techniques
would’ve been used to that of the study reported here. Comparisons can therefore not
be made among these studies. Did any of these studies report on community
analysis?

24. Lines 215-216: ‘Micromonospora’ is mentioned twice.

25. Lines 229-230: See comment above drawing comparisons.

26. The aspect that should be highlighted in the discussion section, is the value of
the different media types in accessing different genera/species. Are there any
reasons why certain genera were isolated on specific media? Any correlation to the
physicochemical properties of the sediment samples?

27. Please make sure that the manuscript is edited for grammar and language usage.
Some examples are listed above, but this is not a comprehensive evaluation.

Reviewer #2: The manuscript (MS) by Guan and colleagues describes the isolation of
novel bacteria from a hypersaline lake in China. The MS is interesting but IMHO
contains several flaws that require attention before it can be accepted for
publication as I point below.

General comments.

1. During the MS when referring to the name and/or lists of different genera, the
authours sometimes use an alphabetical order and sometimes they don’t. They should
be more homogenous through the entire MS. For instance, the abstract is not in
alphabetical order but in lines 172 to 174 the names of the genera mentioned are
precisely in alphabetical oder. It is nt critical to use one or another but
homogeneity would definitively be preferred.

2. It should be clear for the authors that not every reader ifs familiar with the
geography of China. Hence, I would stongly suggest that for the final versión of the
MS, the authours include a map of China showing the exact location of the lake. This
can be referred in section 2.1 Site descriptions and sample collection. In addition,
is there any reference for the sentence “The high salinity, low nutrients level, dry
climate, and high UV intensity makes it an extreme environment.” (Lines 69-70). If
there is any, then it should be added to the MS.

3. Section 2.2. There is a bunch an wide range of seletive isolation media and the
authors claim that they used 9 (line 86, Table 1). However when checking Table 1,
all but one (ie. eight) are media “From this study”. This is intriguing because when
other academics might check the MS in it final form they can argue why and/or how
these eight media were chosen since this is not included nor mentioned anywhere in
the current form of the MS. Therefore, I believe that the authours should add a line
and/or sentence and/or paragraph indicating why and how these media was chosen for
the study.

4. It is not clear to me why there are only 70 sequences deposited in GenBank
(MK818765-MK818834 = 69 sequences plus MK296404 = 70) if the authors mentioned that
they isolated 343 bacteria. That is only a mere 20%. If the identifiction of the
bacteria is purely based on 16S rDNA gene sequencing then I think they should, at
least, submit 50-60% of their sequences to GenBank and not saying that there is
diversity but not providing the corresponding sequences for their isolates.

5. Section 3.1. According to the results from the authors, they sampled in three
different places and provide the information in table 2. In the MS they also
indicate that the conditions of each sample are different and this is very well
exemplified by the high bacterial diversity. Since the authors employed 9 different
media in the study, then I would strongly suggest a table indicating the number of
isolates recovered from each sediment. Perhaps there is one sediment that showed a
higher degree of diversity over the others. These would also help the authors to
decide how many other sequences can be submitted to GenBank rather than submitting
all of them as I suggested for point number 4.

6. Also, how did the authors checked for “duplicating strains”? Again, is this only
base don the 16S rDNA gene sequencing of the isolates? This is indicted in Section
3.2, lines 127-128.

7. I am not convinced by the fact that a 97% similarity (less) refers to a different
species. There are examples of either a novel species or a putative novel genera.
This should be handle with care. If they haven’t done so, then I suggest that the
authors also genome sequence some of the most interesting isolates (ie the ones that
are the most different ones acoording to EZBiocloud) and then make or at least
mention DNA-DNA in silico (available from the DSMZ website) to highlight this point.
However I would strongly object adding the full genome sequencing information to
this MS or it would definitively become a never ending story.

8. Section 3.2 lines 143 – 148. If the reader is not familiar with families and
genera, these lines become extremely confusing because there is n indication of
which genera is either Actinobacteria, Firmicutes, Proteobacteria or
Rhodothermaeota. This is definitively a very interesting paragraph because it
certainly highlights the diversity found in the study but there is no separation
between any phyla thus making it difficult and confusing.

Due to this number of general comments, then I find the Discussion section hard to
follow. Besides, the English should definitively be improved in order to exploit the
full potential of this MS as there are not too many studies on extreme lake
environments.

I should mention that I find the MS very interesting but I am afraid in its present
form I would be against its publication mostly because of its current form. It can
certainly hav a higher impact if it is re-arranged/presented in a better way and
hopefully the authors would be willing to incorporate all of the comments.

Minor comments.

1. Section 2.3, line 97. I believe reference 25 must be the one for the EZBiocloud
website. Apparently some references are misplaced causing confusion because also
reference 26 is not the one for the sentence.

2. Section 2.3, line 102. How many base pairs were obtained for the 16S rDNA gene
sequence? This is not indicated or are the authors assuming that anyone will check
the size of the sequences deposited in the GenBank database?

3. Line 231, shouldn’t it be “are novel” instead of “as novel”?

6. PLOS authors have the option to publish the peer
review history of their article (what does this mean?). If published, this will
include your full peer review and any attached files.

If you choose “no”, your identity will remain anonymous but your review may still be
made public.

**Do you want your identity to be public for this peer review?** For
information about this choice, including consent withdrawal, please see our
Privacy Policy.

Reviewer #1: No

Reviewer #2: No

---

## [Author Response · Author response to Decision Letter 0]

6 Feb 2020

Dear Agnes Pap,

Thank you so much for your kind help. We have replied them item by item based on your
comments. According to the actual needs, we modified the relevant content submitted
online. The sequences of the bacterial isolates reported in this study have been
deposited to GenBank (Accession no. MK818765- MK818834, MK296404). And the Table 5
in our text have been added the revised MS (L214, L218). We hope our reply is
satisfactory. 

We sincerely hope that our paper can be published in PLOS ONE. If you need any more
information concerning the manuscript, please don’t hesitate to contact with us by
e-mail.

Best Wishes

Tong-Wei Guan

to Reviewers 2.6.docx
---

## [Decision Letter · Decision Letter 1]

29 Jun 2020

Isolation and Diversity of Sediment Bacteria in the Hypersaline Aiding Lake,
China

PONE-D-19-20101R1

Dear Dr. Guan,

We’re pleased to inform you that your manuscript has been judged scientifically
suitable for publication and will be formally accepted for publication once it meets
all outstanding technical requirements.

Kind regards,

Paula V Morais, Ph.D

Academic Editor

PLOS ONE

Additional Editor Comments (optional):

Reviewers' comments:

Reviewer's Responses to Questions

**Comments to the Author**

1. If the authors have adequately addressed your comments raised in a previous round
of review and you feel that this manuscript is now acceptable for publication, you
may indicate that here to bypass the “Comments to the Author” section, enter your
conflict of interest statement in the “Confidential to Editor” section, and submit
your "Accept" recommendation.

Reviewer #1: All comments have been addressed

2. Is the manuscript technically sound, and do the data
support the conclusions?

Reviewer #1: Yes

3. Has the statistical analysis been performed
appropriately and rigorously? 

Reviewer #1: Yes

4. Have the authors made all data underlying the
findings in their manuscript fully available?

Reviewer #1: Yes

5. Is the manuscript presented in an intelligible
fashion and written in standard English?

Reviewer #1: Yes

6. Review Comments to the Author

Reviewer #1: (No Response)

7. PLOS authors have the option to publish the peer
review history of their article (what does this mean?). If published, this will
include your full peer review and any attached files.

If you choose “no”, your identity will remain anonymous but your review may still be
made public.

**Do you want your identity to be public for this peer review?** For
information about this choice, including consent withdrawal, please see our
Privacy Policy.

Reviewer #1: No

---

## [Editor Report · Acceptance letter]

1 Jul 2020

PONE-D-19-20101R1 

Isolation and Diversity of Sediment Bacteria in the Hypersaline Aiding Lake, China 

Dear Dr. Guan:

I'm pleased to inform you that your manuscript has been deemed suitable for
publication in PLOS ONE. Congratulations! Your manuscript is now with our production
department. 

Kind regards, 

on behalf of

Prof. Paula V Morais 

Academic Editor

PLOS ONE